# Expression of the Selected Proteins of JAK/STAT Signaling Pathway in Diseases with Oral Mucosa Involvement

**DOI:** 10.3390/ijms24010323

**Published:** 2022-12-24

**Authors:** Kamila Ociepa, Marian Danilewicz, Małgorzata Wągrowska-Danilewicz, Róża Peterson-Jęckowska, Angelika Wójcicka-Rubin, Natalia Lewkowicz, Radosław Zajdel, Agnieszka Żebrowska

**Affiliations:** 1Department of Dermatology and Venereology, Medical University of Lodz, 90-674 Lodz, Poland; 2Department of Pathomorphology, Medical University of Lodz, 92-213 Lodz, Poland; 3Department of Nephropathology, Medical University of Lodz, 92-213 Lodz, Poland; 4Department of Periodontology and Oral Mucosa Diseases, Medical University of Lodz, 92-213 Lodz, Poland; 5Department of Computer Science in Economics, Faculty of Economics and Sociology, University of Lodz, 90-255 Lodz, Poland

**Keywords:** JAK/STAT, pemphigus, pemphigoid, lichen planus, CUS

## Abstract

The JAK/STAT signal pathway is a system of intracellular proteins used by many cytokines and growth factors to express genes responsible for the process of cell activation, proliferation and differentiation. There has been numerous inflammatory and autoimmune diseases identified where the JAK/STAT signaling is disrupted; however, there are only a few papers concerning autoimmune bullous diseases published. The aim of this study was to evaluate the expression of proteins: JAK3, STAT2, STAT4 and STAT6 in epithelium lesions in patients with pemphigus vulgaris (PV), bullous pemphigoid (BP), oral lichen planus (LP) and chronic ulcerative stomatitis (CUS), as well as in the control group. Immunohistochemistry and immunoblotting were used to evaluate expression of selected proteins. We found significantly higher expression of selected JAK/STAT proteins in oral mucosa lesions in study groups in comparison to the control group, which indicates participation of JAK/STAT pathway in pathogenesis of these diseases. In BP and PV there were no increased STAT2 expression, whereas in CUS and LP no increased STAT4 expression occurred. The differences in expression of JAK/STAT proteins in selected disorders have been observed. These results create new potential therapeutic targets for the treatment.

## 1. Introduction

The Janus kinases (JAK) and signal transducers and activators of transcription (STAT) are a group of proteins constituting the signaling pathway present in cells of animals.

The JAK/STAT signal pathway is used by many cytokines and growth factors to express genes responsible for the process of cell activation, proliferation and differentiation. It is a basic relay system in the population of circulating monocytes and lymphocytes involving many cytokines and their receptors [1]. Recent studies seem to indicate an important role of Janus kinases in the pathogenesis of autoimmune skin disorders, with TNF-α, IL-6, IL-4, IFNs, and IL-17 being key mediators for the Th2 cell differentiation [1,2].

However, so far there has been no data published on the role of this pathway in selected disorders. The study concerns pemphigus vulgaris (PV), bullous pemphigoid (BP) lichen planus (LP) and chronic ulcerative stomatitis (CUS) with a similar clinical picture, including blisters and erosions present within both mucous membranes and skin. In the first phase of the disease, only the oral cavity can be affected. The differential diagnosis and treatment of diseases with involvement of oral mucosa is a major challenge in clinical practice.

That is why the aim of this study was to evaluate the expression of proteins: JAK3, STAT2, STAT4 and STAT6 in epithelium lesions in patients with PV, BP, LP and CUS, as well as in the control group. We would like to indicate the JAK/STAT signaling pathway proteins that may constitute a marker of selected autoimmune mucosal diseases in their active phase and the target for the future treatment.

## 2. Results

### 2.1. Immunohistochemistry

The immunoreactivity of JAK3 and STAT4 antibodies was detected in several cells in the oral mucosa in the control group (C). Expression of JAK3, STAT2, STAT4 and STAT6 was cytoplasmic and membranous.

In the STAT2 expression in pemphigus, we did not find a statistically significant difference (*p* = 0.58) compared to the control group. In pemphigoid the result was not statistically significant (*p* = 0.38) as compared to the control group. The expression of STAT2 protein was significantly higher in CUS patients (*p* < 0.04) and LP patients (*p* < 0.02) as compared to the healthy volunteers (Figure 1 and Table 1).

The expression of STAT4 protein was significantly higher in PV (*p* = 0.006) and BP patients (*p* = 0.001) as compared to the C. There was no statistical difference in expression of STAT4 in CUS (*p* < 0.06) and LP (*p* = 0.02) patients as compared to the C (Figure 1 and Table 2).

The expression of STAT6 protein was significantly higher in PV (*p* = 0.04), BP (*p* = 0.02) and LP patients (*p* = 0.05) as compared to the C. There was no statistical difference in expression of STAT6 in CUS patients (*p* < 0.49) as compared to the C (Figure 2 and Table 3).

Expression of JAK3 protein was higher in PV (*p* < 0.02), BP (*p* < 0.007), LP (*p* < 0.04) and CUS patients (*p* < 0.03) as compared to the C (Figure 3 and Table 4).

### 2.2. Immunoblotting

Immunoblotting confirmed results of the expression of selected proteins of JAK/STAT pathway in immunochemistry method.

The intensity of JAK3 expression was higher in all study groups (PV: 133.48 ± 0.84, *p* < 0.05; BP: 132.21 ± 0.96; *p* < 0.05; CUS: 134 ± 1.03 *p* < 0.05; LP: 132 ± 0.89 *p* < 0.05) as compared to the control group (130.42 ± 1.65; *p* < 0.05) (Figure 4).

The expression of STAT2 was evaluated higher in CUS patients (145.83 ± 0.25) and LP patients (143.85 ± 3.09) as compared to the control group (136.28 ± 2.84; *p* < 0.05). There was no significant difference between STAT2 expression in PV and BP patients vs. the control group (Figure 5).

The intensity of STAT4 expression was higher in PV patients (138.39 ± 0.84) and BP patients (141.2 ± 0.05) as compared to the control group (121.63 ± 1.75; *p* < 0.05). There was also statistical significance between expression of STAT4 in CUS and LP patients (Figure 6).

The expression of STAT6 protein was significantly higher in PV patients (131.37 ± 2.55), BP patients (129.34 ± 1.37) and LP (130.56 ± 1.76) as compared to the control group (123.48 ± 1.13; *p* < 0.05). There was no statistical difference in the expression of STAT6 in CUS patients vs. control group (Figure 7).

## 3. Discussion

Nishio et al. have confirmed [3] the presence of the JAK/STAT signal pathway proteins in healthy human epidermis. These data may indicate that the proper functioning and maintenance of the immune balance of healthy epidermis may depend on the presence of a certain level of JAK/STAT expression in its structures [3]. By analogy, the JAK/STAT signal path probably plays a role in the proper functioning of epithelial cells of the mucosa oral cavity.

Our studies assessing the expression of JAK3, STAT2, STAT4 and STAT6 in normal sections of the oral mucosa prove that a certain constitutive expression of this pathway is present in the healthy epithelium.

Many recent studies have indicated involvement of the JAK/STAT cytokine signaling mechanism in the pathogenesis of inflammatory diseases [1,2]. T lymphocyte-mediated inflammation with high level of proinflammatory cytokines is well recognized in many dermatoses. Based on this data, it seems reasonable to suspect that this pathway may play a role in the pathogenesis of skin disorders such as BP, PV [2], as well as in LP and CUS.

In PV, the majority of studies have suggested a shift in the balance towards Th2 with elevated levels of IL-4, IL-6, and IL-10 in serum [4,5,6]. TNF- α, IL-6, IL-15, IL-10, IL-12, are involved in the mechanism of pemphigus lesions formation [6,7]. There are studies showing both elevated and decreased levels of IFN-γ and IL-2, as well as studies in which these cytokines are at the same level in the serum of individuals with PV compared to healthy individuals [5]. Certainly, there is a need for further research on the issue of Th1/Th2 imbalance in the etiopathogenesis of PV.

However, the presence in our study of increased JAK3 expression proves that these mechanisms play a role in the pathogenesis of PV [5]. JAK3 is involved in the signaling of IL-2, which is probably responsible for the recurrence of lesions in the course of PV [8]. This allows us to think that JAK3 could act as a marker for monitoring the activity of the disease. Elevated levels of IL-2 in the serum of patients with PV may confirm the role of the examined pathway in the pathomechanism of skin lesions in the course of PV [6].

Studies on proinflammatory cytokines largely show elevated TNF-α and IL-6 levels in the serum of PV patients [9]; several studies have also reported elevated levels of IL-1 in the lesional tissue [10]. There are also reports of elevated IL-8 in bullous fluid and serum in PV patients [11]. Also, in the etiopathogenesis of PV an important role is attributed to IL-12. In the study of Masjedi et al. [12], there was a significantly elevated serum IL-12 concentration in PV patients [12].

In our studies, we found increased expression of STAT4 in PV. The main activator of STAT4 is IL-12. STAT4 signaling initiated by IL-12, IFN-α and IFN-β is essential for the differentiation of the Th1 cell line. At the same time, it plays a key role in the production of IFN-γ in response to IL-12 [13].

In recent studies, changes in T and B cells populations dysfunction in PV have been proven. The B cells are strongly stimulated to produce antibodies through a number of cytokines secreted by Th2 cells [14]. In our studies we have found a high level of STAT6 expression in the PV lesions, opposite of the control group. These results suggest that STAT6 plays a role in the etiopathogenesis of PV, most likely by mediating a Th2-dependent cellular response.

The STAT6 protein is activated by IL-4 and IL-13. According to published data, in the serum of patients with PV there is an elevated level of IL-4 [14]. This seems to be why we obtained elevated STAT 6 levels in patients with PV.

Cytokines, inflammatory mediators and their binding to JAK/STAT signal pathway proteins play a role in the pathogenesis of BP. The role of Th2 lymphocytes in the pathogenesis of BP is demonstrated by the presence of elevated levels of Th2 cytokines: IL-4, IL-5, IL-6 and the soluble receptor for IL-2 [14,15,16]. Engman et al. [17] showed that an essential element of blister formation is early skin infiltration by activated CD4+ T lymphocytes and eosinophils [17].

This confirms our research, in which the results of increased JAK3 expression in lesions of BP patients were obtained. By analyzing the profile of cytokines involved in the pathomechanism of BP and cytokines involved in the JAK3-dependent signaling pathway, one can observe their correlation. JAK3 is the signaling pathway for IL-2, IL-4, IL-7, IL-9, IL-15 and IL-21 cytokines [18,19]. This proves the involvement of JAK3 protein in the etiopathogenesis of BP. In addition, activation of JAK3 is closely related to IL-2, and elevated concentrations of Il-2 also were confirmed in BP [18,19].

In our study, increased expression of STAT4 protein was observed on the oral mucosa in patients with BP. The role of STAT4 protein in the etiopathogenesis of BP is probably related to the activation of STAT2 protein in the Th1-dependent response. The same conclusions can be drawn from the study by Juczynska et al. [2] which confirmed the increased expression of STAT4 in skin lesions of BP patients [2].

It is now known that STAT6 is activated by two cytokines: IL-4 and IL-13 [20]. Rico et al. [14] has shown an increased level of IL-4, IL-5 and IL-13 in pemphigoid lesions. Our results indicating increased expression of this pathway protein indirectly indicate a role of IL-4 in the pathomechanism of BP.

The increase of STAT6 expression in biopsies of BP patients can also be explained by both the involvement of the Th2 response in the pathogenesis of pemphigoid and the specific eosinophilic cytokine profile present in this disease entity [15].

In patients with lichen planus the percentage of peripheral Th1 and Th17 cells is much higher than in the healthy volunteers [21]. On the basis of many studies, it was confirmed that Th17 cells produce proinflammatory cytokines such as: IL-17, IL-17F, IL-21 and IL-26. Cytokines involved in the Th1 and Th17 dependent response in signal transduction use the JAK/STAT path [22].

The JAK3 expression results are consistent with the earlier assumptions that the increase in JAK3 protein expression takes place primarily in T lymphocytes, which mediate inflammatory reactions that play a key role in the etiopathogenesis of LP, as well as described PV and BP.

In LP, we observe a cytotoxic reaction that leads to the release of a number of cytokines: IL-1, IL-3, IL-6, IL-8, TNF-α and INF-γ. CD8 + cells present in infiltrates in the skin and mucous membranes of LP are responsible for the cytolysis of the basal layer [23,24].

We showed significantly increased expression of STAT2 and STAT6 in lesions in LP patients. The STAT2 is involved in the response mediated by IFN-α and IFN-β. The activity of the STAT2 protein is often closely related to the activity of the STAT1 protein. The activation of STAT2-STAT1 is related to each other, in a mechanism in which IFN-α activates STAT1 on a STAT2-dependent pathway [25]. Taken to consideration of STAT2 and STAT6 dependent cytokines, further studies on the participation of inflammatory mediators in the induction of lesions in LP are important.

In biopsies of CUS patients we found increased expression of STAT6 and STAT2. These results may indicate the similarity of pathomechanisms occurring during the formation of erosions in CUS, to those that have been described in LP. We can expect that the participation of IFN-α and IFN-β will be necessary for the development of CUS. There is no literature data about this topic.

In CUS, STAT6 expression was comparable to the results obtained in the control group. This seems to indicate an advantage of Th1 response mechanisms in the pathogenesis of CUS [26]. Conducting further research is necessary to understand the immunological mechanisms involved in the development of CUS lesions.

## 4. Materials and Methods

### 4.1. Patients

For the study, a group of 181 patients from the Department of Periodontology and Oral Implantology and the Department of Dermatology and Venereology at the Medical University of Lodz, was initially selected. Patients enrolled in the study were diagnosed during the active phase of the disease, with the presence of active erosions and blisters in the oral cavity at the beginning of the diagnosis. At the time of collecting the material for the study, they were not receiving any treatment (neither systemic nor topical). The study protocol RNN/132/07/KB was approved by the Local Ethical Committee of the Medical University of Lodz.

Finally, the research group consisted of 112 patients with confirmed diagnosis: 28 with PV and a mean age of 54.4; 31 with BP and a mean age of 67.3; 38 with LP and a mean age of 52.8; and 15 with CUS and a mean age of 64.6. The control group comprised 25 healthy volunteers with a mean age of 47.3 (Table 5).

The diagnosis of PV, BP, LP and CUS was established based on medical history, clinical picture and immunofluorescence findings. In all PV patients, the presence of IgG network-like deposits were found in the direct immunofluorescence test (DIF) (Euroimmun, Lübeck, Germany) study. The positive indirect immunofluorescence method (IIF, Euroimmun, Lübeck, Germany) on the monkey esophagus substrate was demonstrated in the whole group. In 12 patients, IIF was positive at 1:40 titer and the rest of the patients had positive IIF at 1:320 dilutions. Circulating antibodies against Dsg3 were detected in the whole group in the DSG-3 ELISA test (Euroimmun, Lübeck, Germany).

In all BP patients, the presence of linear IgG and/or C3 component deposits along the BMZ was found in the DIF (Euroimmun, Lübeck, Germany). The salt split test was positive for epidermal part of artificial blister in all BP cases. The IIF on the monkey esophagus substrate showed circulating IgG antibodies against basement membrane zone (Euroimmun, Lübeck, Germany) in 23/31 patients, in titer range from 1:80 to 1:160. For serum 19 of 31 patients demonstrated the presence of anti BP180NC16a autoantibodies (Euroimmun, Lübeck, Germany).

CUS patients confirmed the presence of SES-ANA antibodies (Euroimmun, Lübeck, Germany) in titer range from 1:40 to 1:640, in the IIF.

### 4.2. Methods

Immunohistochemical and immunoblotting methods were used to evaluate the expression of the examined proteins in lesional epithelium.

#### 4.2.1. Immunohistochemistry

Paraffin-embedded tissue sections were mounted onto Superfrost slides, deparaffinized, then treated in a solution of TRS (Target Retrieval Solution, Dako, Glostrup, Denmark) and transferred to distilled water. Endogenous peroxidase activity was blocked and then sections were rinsed with Tris-buffered saline (TBS, Dako, Glostrup, Denmark), and incubated with primary rabbit polyclonal antibody against STAT2 mouse monoclonal antibody against STAT4 (primary rabbit polyclonal antibody against STAT6 and incubated overnight with mouse monoclonal antibody against JAK3 (Santa Cruz biotechnology Inc, Dallas, TX, USA). Immunoreactive proteins were visualized using EnVision-horseradish peroxidase kit (Dako, Carpinteria, CA, USA). After washing, the sections were counter-stained with hematoxylin and cover slipped. For each antibody and for each sample a negative control were processed. The expression of STAT2, STAT4, and STAT6 are expected to be cytoplasmatic and nuclear upon activation.

#### 4.2.2. Semiquantitative Analysis

In each specimen staining intensity of JAK3, STAT2, STAT4, and STAT6 were recorded semi quantitatively in 5–7 high power fields by two independent observers. The tissue sections were scored based of the percentage of immunostained cells: 0% to 10% = 0, 10% to 30% = 1, 30% to 50% = 2, 50% to 70% = 3 and 70% to 100% = 4. Sections were also scored on the basis of staining intensity: negative = 0, mild = 1, moderate = 2 and intense = 3.

#### 4.2.3. Immunoblotting

The Western blot method was used to evaluate the expression of JAK3, STAT2, STAT4 and STAT6. Total protein from frozen skin samples from PV, BP, CUS and LP patients and healthy controls were extracted in RIPA protein extraction buffer, supplemented with protease inhibitor cocktail (Sigma-Aldrich, St. Louis, MO, USA). The lysate was centrifuged and the pellet was discarded. Protein concentrations were determined by the BCA Protein Assay Kit (Pierce Thermo Scientific, Waltham, MA, USA) according to manufacturer’s instructions.

The membrane was blocked with non-fat milk in TBST and then incubated with the mouse primary antibodies (Santa Cruz Biotechnology, Dallas, TX, USA). Afterwards, the membrane was incubated with secondary goat anti-mouse IgG polyclonal antibodies conjugated with alkaline phosphatase (Santa Cruz Biotechnology, Dallas, TX, USA). The bands were developed using BCIP/NBT Alkaline Phosphatase Substrate (Merck Millipore, Darmstadt, Germany), analyzed using the ImageJ 1.34s software (Wayne Rasband, National Institutes of Health, Bethesda, MD, USA), which allowed for image analysis of densitometry expressed as % optical density (OD) over the background. Obtained results were expressed as the mean ± SD.

#### 4.2.4. Statistical Methods

The results were presented as the mean ±  SEM. The data were analyzed using Statistica (v. 10.0; StatSoft, Tulsa, OK, USA). The distribution of the data and the equality of variances were checked by Levene’s test. Differences between groups were tested using ANOVA (WB), unpaired Student’s *t*-test (immunohistochemistry) and The Mann–Whitney U test was used where appropriate. The level of significance was defined where *p* < 0.05.

## 5. Conclusions

It seems that disturbances in the balance between Th1/Th2 lymphocyte subpopulations play an important role in the pathomechanism of all studied disorders. We showed significantly increased JAK3 expression in biopsies of mucosal lesions in all selected diseases as compared to the control group.

The JAK3 expression was significantly higher in the PV and BP patients. This provides the basis for planning further studies of JAK3 expression in pathogenesis of PV and BP. Despite the lack of studies on the participation of particular groups of lymphocytes in the course of CUS, based on the result of increased JAK3 expression in CUS patients, we can assume that disturbances between Th1/Th2 response will play a role in the pathogenesis of this disease.

In our study the differences in STAT expression in lesions of selected diseases have been observed. In BP and PV there was no increased STAT2 expression, whereas in CUS and LP no increased STAT4 expression occurred. These results suggest other mechanisms involved in the pathomechanism of lesions in LP and CUS as compared to autoimmune blistering diseases such as PV and BP. We hope that the differences in the expression of these proteins could be markers of specific disease entities to target for future treatment.

The JAK/STAT pathway is an new area of interest in dermatology, but further studies are required. Conducted studies to date suggest that the pathway may be connected with inflammatory dermatoses, and Janus kinases path analysis may have a place in future disease treatment. The creation of alternative strategies for the treatment of autoimmune diseases is necessary due to still imperfect traditional therapy and complications after them. JAK/STAT pathway inhibitors are a promising therapeutic regimen.

## Figures and Tables

**Figure 1 ijms-24-00323-f001:**
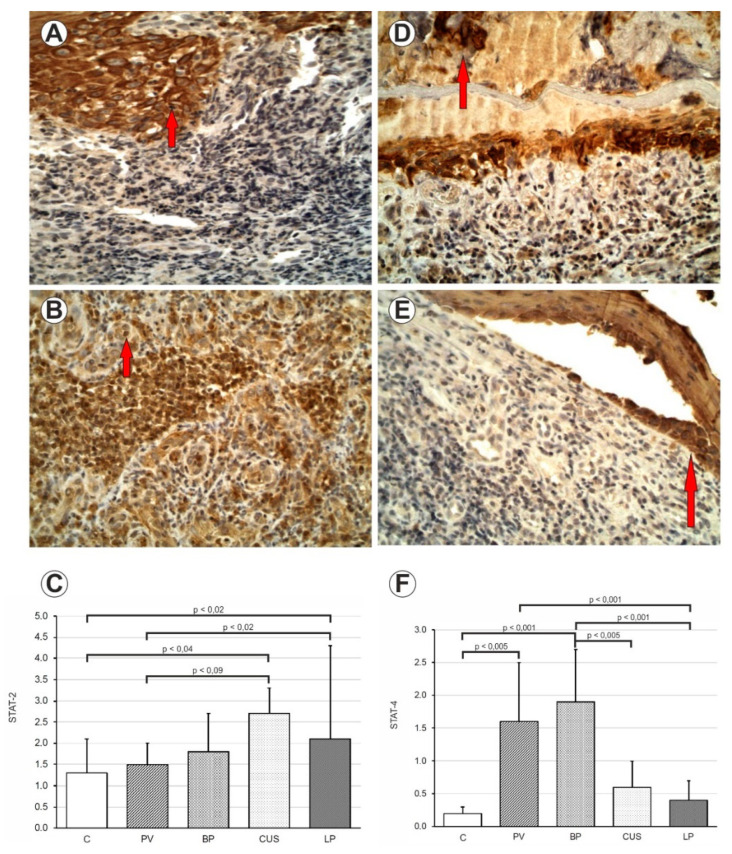
(**A**) Immunohistochemistry. STAT2 expression in epithelial cells and inflammatory influx. CUS. 400× mag. (**B**) Immunohistochemistry. STAT2 expression in epithelial cells and inflammatory influx. LP. 400× mag. (**C**) Comparison of STAT2 expression in PV, BP, CUS, LP and healthy mucosa (**C**). (**D**) Immunohistochemistry. STAT4 expression in epithelial cells and inflammatory influx. PV 400× mag. (**E**) Immunohistochemistry. STAT4 expression in epithelial cells and inflammatory influx. BP 200× mag. (**F**) Comparison of STAT4 expression in PV, BP, CUS, LP and healthy mucosa (**C**).

**Figure 2 ijms-24-00323-f002:**
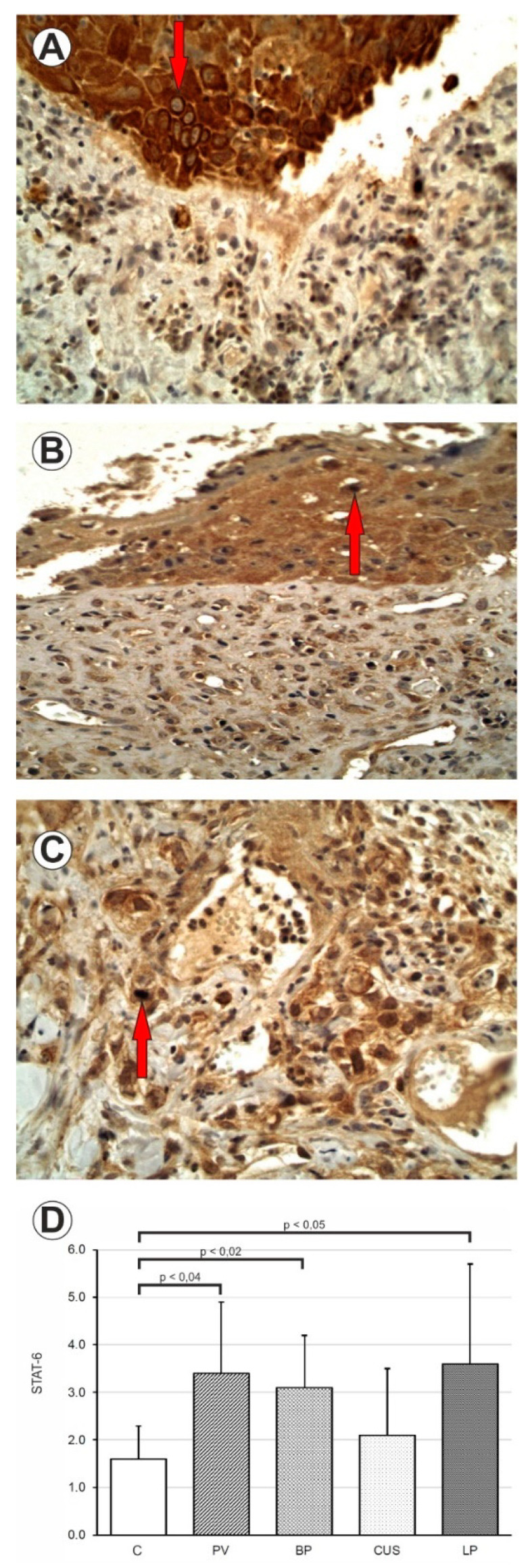
(**A**) Immunohistochemistry. STAT6 expression in epithelial cells and inflammatory influx. PV. 400× mag. (**B**) Immunohistochemistry. STAT6 expression in epithelial cells and inflammatory influx. BP. 400× mag. (**C**) Immunohistochemistry. STAT6 expression in epithelial cells and inflammatory influx. LP. 400× mag. (**D**) Comparison of STAT6 in PV, BP, CUS, LP and healthy mucosa (**C**).

**Figure 3 ijms-24-00323-f003:**
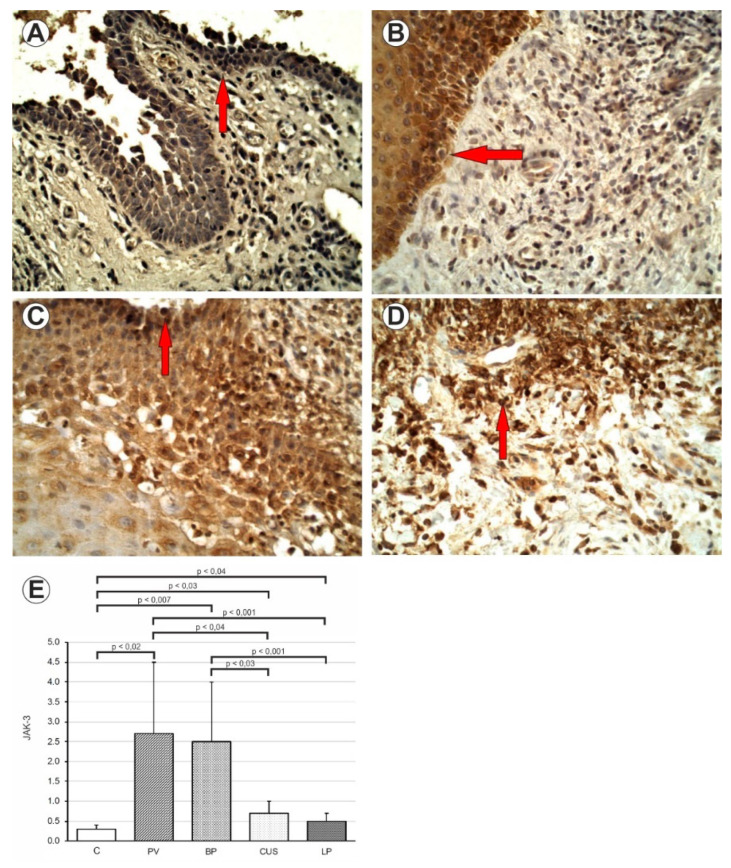
(**A**) Immunohistochemistry. JAK3 expression in epithelial cells and inflammatory influx. PV. 200× mag. (**B**) Immunohistochemistry. JAK3 expression in epithelial cells and inflammatory influx. BB. 400× mag. (**C**) Immunohistochemistry. JAK3 expression in epithelial cells and inflammatory influx. CUS. 400× mag. (**D**) Immunohistochemistry. JAK3 expression in epithelial cells and inflammatory influx. LP. 400× mag. (**E**) Comparison of JAK3 in PV, BP, CUS, LP and healthy mucosa (**C**).

**Figure 4 ijms-24-00323-f004:**
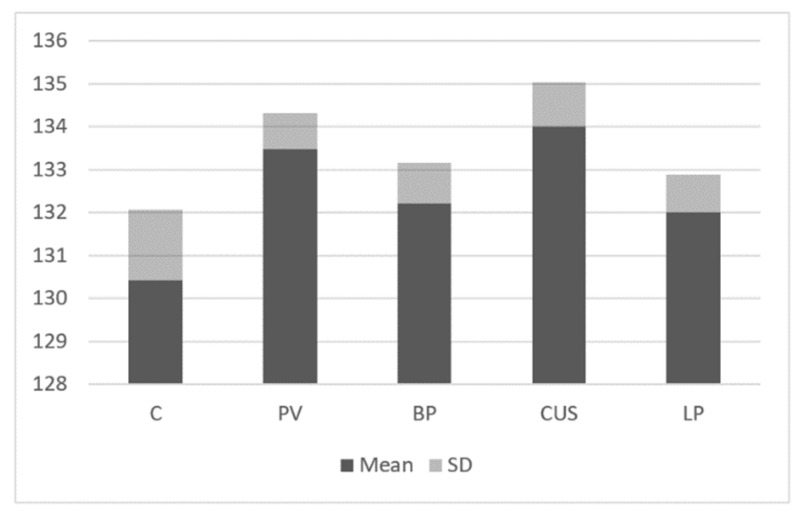
Immunoblotting. JAK3 expression in study groups and control group.

**Figure 5 ijms-24-00323-f005:**
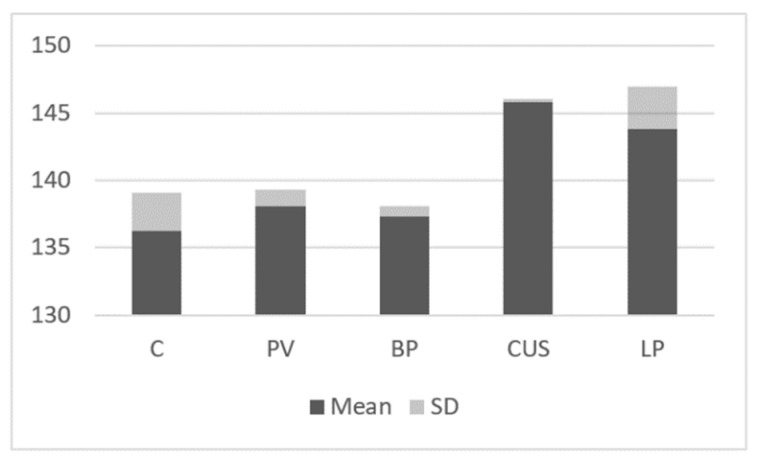
Immunoblotting.STAT2 expression in studies groups and control group.

**Figure 6 ijms-24-00323-f006:**
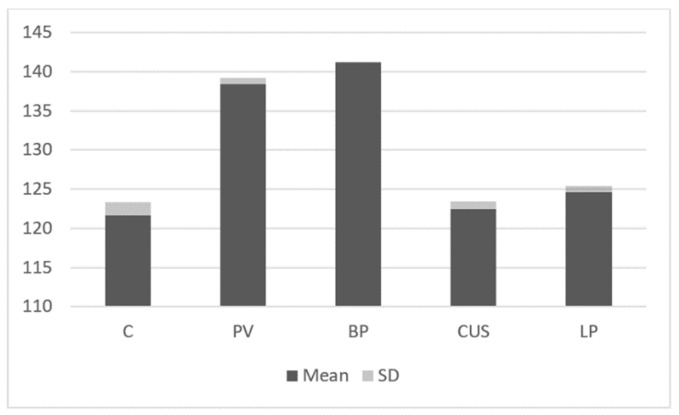
Immunoblotting. STAT4 expression in studies groups and control group.

**Figure 7 ijms-24-00323-f007:**
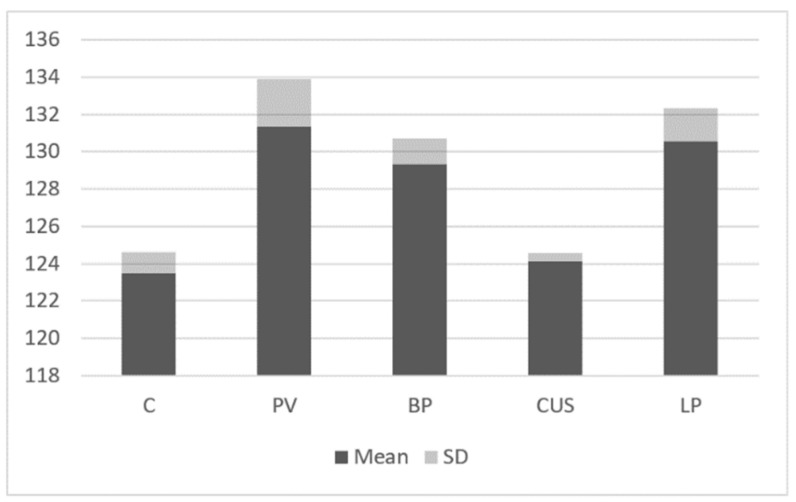
Immunoblotting. STAT6 expression in studies groups and control group.

**Table 1 ijms-24-00323-t001:** Expression of STAT2 in study groups. C—control group; BP—pemphigoid; PV—pemphigus vulgaris; LP—lichen planus; CUS—chronic ulcerative stomatitis.

C vs. PV *p* = 0.58 NS	C vs. BP *p* = 0.38 NS	C vs. CUS *p* < 0.04	C vs. LP *p* < 0.02
PV vs. BP *p* = 0.48, NS	PV vs. CUS *p* < 0.09	PV vs. LP *p* < 0.02	
BP vs. LP *p* = 0.23 NS	BP vs. CUS *p* = 0.13 NS	CUS vs. LP *p* = 0.05 NS	

**Table 2 ijms-24-00323-t002:** Expression of STAT4 in study groups. C—control group; BP—pemphigoid; PV—pemphigus vulgaris; LP—lichen planus; CUS—chronic ulcerative stomatitis.

C vs. PV *p* < 0.006	C vs. BP *p* < 0.001	C vs. CUS *p* = 0.06 NS	C vs. LP *p* = 0.2 NS
PV vs. BP *p* = 0.45 NS	PV vs. CUS *p* < 0.05	PV vs. LP *p* < 0.001	
BP vs. LP *p* < 0.001	BP vs. CUS *p* < 0.005	CUS vs. LP *p* = 0.18 NS	

**Table 3 ijms-24-00323-t003:** Expression of STAT6 in study groups. C—control group; BP—pemphigoid; PV—pemphigus vulgaris; LP—lichen planus; CUS—chronic ulcerative stomatitis.

C vs. PV *p* < 0.04	C vs. BP *p* < 0.02	C vs. CUS *p* = 0.49 NS	C vs. LP *p* < 0.05
PV vs. BP *p* = 0.62 NS	PV vs. CUS *p* = 0.14 NS	PV vs. LP *p* = 0.78 NS	
BP vs. LP *p* = 0.45 NS	BP vs. CUS *p* = 0.14 NS	CUS vs. LP *p* = 0.13 NS	

**Table 4 ijms-24-00323-t004:** Expression of JAK3 in study groups. C—control group; BP—pemphigoid; PV—pemphigus vulgaris; LP—lichen planus; CUS—chronic ulcerative stomatitis.

C vs. PV *p* < 0.02	C vs. BP *p* < 0.007	C vs. CUS *p* < 0.03	C vs. LP *p* < 0.04
PV vs. BP *p* = 0.7 NS	PV vs. CUS *p* < 0.04	PV vs. LP *p* < 0.001	
BP vs. LP *p* < 0.001	BP vs. CUS *p* < 0.03	CUS vs. LP *p* = 0.05 NS	

**Table 5 ijms-24-00323-t005:** Demographic data and immunohistopathologic findings (increased expression of selected protein vs. control group). PV—pemphigus vulgaris; BP—pemphigoid; LP—lichen planus; CUS—chronic ulcerative stomatitis; and C—control group.

	Number of Patients	Age (Years)	Immunopathologic Results	STAT2	STAT4	STAT6	JAK3
PV	28	54.4 (45–64)	28/28 anti Dsg3 (1:40–1:320)	-	+	+	+
BP	31	67.3 (46–89)	23/31 anti BMZ (1:80–1:160) 19/31 anti NC16a	-	+	+	+
LP	38	64.6 (51–82)	Negative	+	-	+	+
CUS	15	47.3 (22–61)	SES ANA (1:40–1:640)	+	-	-	+
C	25	54.4 (45–64)	Negative	-	-	-	-

## Data Availability

Not applicable.

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
