# Peer review of "Expression of the Selected Proteins of JAK/STAT Signaling Pathway in Diseases with Oral Mucosa Involvement"

_ijms, 2022, doi:10.3390/ijms24010323_

Round 1

Reviewer 1 Report

Pemphigus and pemphigoid are blistering skin diseases that are easily diagnosed on direct immunofluorescence and histopathology, followed clinically and serologically ( for pemphigus- with dsg1,3 antibody or ICSA levels). Pemphigus is also easily treatable with steroids+mycophenolate if mild or very successfully with low dose Rituximab 1g X2 and patients successfully rendered into a durable remission.

The issues i have with this paper are:

1.Lack of originality and clinical angle to this- ie. how is this going to improve or change diagnosis or management of such patients

2.Lack of clear definitions of treatment or disease activity at time of sample collection- this will affect cytokines in skin.

3.No corresponding cytokine measurement in blood. Perhaps a ratio of tissu resident: blood may be helpful. A ROC curve of these results vs dsg1.3 antibodies may be more practically useful too. Immunoblotting is an archaic method Can use adressable laser bead assays.

4.Wavering and long winded description of JAK/STAT pathway and confusion over which blistering disease is Th1/Th2/Treg driven ,and why.

Author Response

                                                                                           28th November 2021

Dear Reviewers,

We are very grateful for the remarks of the Reviewers. There is no doubt that these reviews helped us to improve our paper. All remarks were applied to the new version of the manuscript and requested corrections were made.

Below we have referred to all the comments of the Reviewers:

Reviewer #1:  

“Pemphigus and pemphigoid are blistering skin diseases that are easily diagnosed on direct immunofluorescence and histopathology, followed clinically and serologically (for pemphigus- with dsg1,3 antibody or ICSA levels). Pemphigus is also easily treatable with steroids+mycophenolate if mild or very successfully with low dose Rituximab 1g X2 and patients successfully rendered into a durable remission. 

The issues I have with this paper are:

1.Lack of originality and clinical angle to this- ie. how is this going to improve or change diagnosis or management of such patients

2.Lack of clear definitions of treatment or disease activity at time of sample collection- this will affect cytokines in skin.

3.No corresponding cytokine measurement in blood. Perhaps a ratio of tissu resident: blood may be helpful. A ROC curve of these results vs dsg1.3 antibodies may be more practically useful too. Immunoblotting is an archaic method Can use adressable laser bead assays.

4.Wavering and long winded description of JAK/STAT pathway and confusion over which blistering disease is Th1/Th2/Treg driven ,and why.”

The differential diagnosis and treatment of diseases with involvement of oral mucosa is a major challenge in clinical practice.The study concerns pemphigus vulgaris (PV), bullous pemphigoid (BP) lichen planus (LP) and chronic ulcerative stomatitis (CUS) with a similar clinical characteristics including the occurrence of bullous lesions and erosions in the oral cavity. Difficulties apply especially when the lesions affect only the oral cavity (without the initial involvement of the rest of the skin).

The overexpression of various proteins of JAK/STAT has been reported in many diseases of different pathogenesis. We have studies on the expression of these proteins in psoriasis, atopic dermatitis and alopecia areata, and now we also have the opportunity to treat these disorders with Janus kinase inhibitors.

However, as so far there has been no data published on the expression of JAK3, STAT2, STAT4 and STAT6 proteins family in the BP, PV, CUS and erosive LP and the pathogenesis of them is still unclear, we are convinced that the current observations are worth publishing and consideration in further studies.

Ad. 1. We would like to indicate of JAK/STAT signaling pathway proteins that may constitute a marker of selected autoimmune mucosal diseases in their active phase and possible target for treatment. Treatment with steroids plus mycophenolate has side effects (especially with high doses of steroids). Rituximab is a very effective treatment but requires an intravenous infusion. JAK/STAT pathway inhibitors are an oral treatment, and therefore are very attractive to our patients. There is still a therapeutic problem in achieving lasting remission in CUS and erosive lichen planus. Unfortunately, as is well known, these diseases tend to be chronic.

Ad. 2. Patients enrolled in the study were diagnosed during the active period of the disease, with the presence of active erosions and blisters in the oral cavity. At the time of collecting the material for the study, they were not receiving any treatment (neither systemic nor topical). We have now described this clearly in page 10, line 230.

Ad.3. Primarily, immunohistochemistry and histological morphometry with a computerized analysis system were utilized to analyze specific expression of JAK3, STAT2, STAT4 and STAT6, simultaneously in epithelial and in influxed inflammatory cells of patients and healthy skin of volunteers. We are aware that the severity of inflammatory influx might influence on the results of immunoblotting. Therefore, performing this technique we intended to confirm only their overexpression in the whole lesion area as compared to perilesional skin and to healthy skin, without differentiation of cell types. What is more, utilizing immunoblotting, we aimed to differentiate the expression of selected proteins between these four different diseases. This technique is still used for this type of research.

In pemphigus, we could correlate the obtained results with the level of anti-Dsg3 and anti-Dsg1 antibodies, in pemphigoid with anti-NC16 A (although they are not present in all patients with BP). Such a juxtaposition with the SES ANA antibodies in the CUS would be troublesome - another detection technique. And in the case of lichen planus patients, there is no such marker at all.

Proteins JAK3, STAT2, STAT4 and STAT6 are involved in action of many types of cytokines. For example JAK3: IL-2, IL-7 IL-4, IL-9, IL-15 and IL-21; STAT2: IFN-α and IFN-β; STAT4: IFN, IL-12, IL-21 and IL-23 and STAT6: IL-4, Il-13 and it would be difficult to verify the serum levels of all these mediators. Undoubtedly, this is an issue for further research.

Nevertheless, we are aware that further studies also including other skin diseases are needed to confirm our results and connection of studied proteins and circulating cytokines.

Ad.4
In the current form of the paper we have shortened and reorganized the whole manuscript, including discussion part.

And the last issue - the English language has been improved according to the comments of both reviewers.

                                                                                                      Kind regards

                                                                                       Agnieszka Å»ebrowska

Reviewer 2 Report

The manuscript entitled “Expression of the selected proteins of JAK/STAT Signaling Pathway in diseases with oral mucosa involvement” written by Kamila Ociepa et al., contains an interesting insight in the involvement of the JAK/STAT signal pathway  in diseases with oral mucosa. However, this article cannot be published in the present form because of some questions, which it arises.

1/English language: The manuscript is still poor written and it deserves a severe revision before resubmission. It is possible that some phrases to be hard to understand, but I am not an English expert. Besides there are many typographic mistakes all around the text.

2/ Authors should use the same abbreviation in all the text such as CG for control group not a “C”

3/Explain abbreviation in the abstract part

4/Conclusion in abstract part should be changed

5/ add titles in the results part

6/ introduction part is so poorly

7/ Conclusion part is too long

8/ Explain the common pathophysiological point between the four pathologies

Author Response

               28th November 2022

Dear Reviewers,

We are very grateful for the remarks of the Reviewers. There is no doubt that these reviews helped us to improve our paper. All remarks were applied to the new version of the manuscript and requested corrections were made.

Reviewer #2:

“The manuscript entitled “Expression of the selected proteins of JAK/STAT Signaling Pathway in diseases with oral mucosa involvement” written by Kamila Ociepa et al., contains an interesting insight in the involvement of the JAK/STAT signal pathway  in diseases with oral mucosa. However, this article cannot be published in the present form because of some questions, which it arises.

1/English language: The manuscript is still poor written and it deserves a severe revision before resubmission. It is possible that some phrases to be hard to understand, but I am not an English expert. Besides there are many typographic mistakes all around the text.

2/ Authors should use the same abbreviation in all the text such as CG for control group not a “C”

3/Explain abbreviation in the abstract part

4/Conclusion in abstract part should be changed

5/ add titles in the results part

6/ introduction part is so poorly

7/ Conclusion part is too long

8/ Explain the common pathophysiological point between the four pathologies.”

We have addressed all of the Reviewer's comments below:

Ad. 1. The English language has been improved according to the comments of both reviewers. We have corrected the entire manuscript. We have removed typographical errors from the text.

Ad. 2. Throughout the text we have used the abbreviation C for the control group. This abbreviation appears on all graphs and in the text.

Ad. 3. We have explained abbreviation in the abstract part.

Ad. 4 We have changed the conclusion in abstract section.

Ad. 5. We have added titles in the result section.

Ad. 6. And Ad. 7. In the current form of the paper we have shortened and reorganized the entire manuscript, including the introduction and the discussion section.

Ad. 8. Following this suggestion, we have improved the discussion section.  In the current form of the paper we have shown the pathophysiological link between the four pathologies.

                                                                                                         Kind regards

                                                                                                Agnieszka Zebrowska

Round 2

Reviewer 2 Report

Accept in this form